# *Solanum nigrum* Fruit Extract Increases Toxicity of Fenitrothion—A Synthetic Insecticide, in the Mealworm Beetle *Tenebrio molitor* Larvae

**DOI:** 10.3390/toxins12100612

**Published:** 2020-09-24

**Authors:** Marta Spochacz, Monika Szymczak, Szymon Chowański, Sabino Aurelio Bufo, Zbigniew Adamski

**Affiliations:** 1Department of Animal Physiology and Development, Institute of Experimental Biology, Faculty of Biology, Adam Mickiewicz University in Poznań, ul. Uniwersytetu Poznańskiego 6, 61-614 Poznań, Poland; monikasz@amu.edu.pl (M.S.); szyymon@amu.edu.pl (S.C.); zbigniew.adamski@amu.edu.pl (Z.A.); 2Department of Sciences, University of Basilicata, Via dell’Ateneo Lucano 10, 85100 Potenza, Italy; sabino.bufo@unibas.it; 3Department of Geography, Environmental Management and Energy Studies, University of Johannesburg, Auckland Park Kingsway Campus, Johannesburg 2092, South Africa; 4Electron and Confocal Microscope Laboratory, Faculty of Biology, Adam Mickiewicz University in Poznań, ul. Uniwersytetu Poznańskiego 6, 61-614 Poznań, Poland

**Keywords:** *Solanum nigrum* extract, fenitrothion, *Tenebrio molitor*, beetles, transmission electron microscopy, glycogen, protein, lipid, fat body, midgut

## Abstract

Synthetic insecticides are widely used for crop protection both in the fields and in the food stored facilities. Due to their toxicity, and assumptions of Integrated Pest Management, we conducted two independent experiments, where we studied the influence of *Solanum nigrum* unripe fruit extract on the toxicity of an organophosphorus insecticide fenitrothion. In the first variant of the experiment, *Tenebrio molitor* larvae were fed with blended fenitrothion (LC_50_) and the extract in four concentrations (0.01, 0.1, 1 and 10%) in ratio 1:1 for 3 days. In the second variant, a two-day application of fenitrothion (LC_40_) was preceded by a one-day extract treatment. The first variant did not show any increase in lethality compared to fenitrothion; however, ultrastructure observations exhibited swollen endoplasmic reticulum (ER) membranes in the midgut and nuclear and cellular membranes in the fat body, after application of blended fenitrothion and extract. An increased amount of heterochromatin in the fat body was observed, too. In the second variant, pre-treatment of the extract increased the lethality of larvae, decreased the level of glycogen and lipids in the fat body and disrupted integrity of midgut cellular membranes. *S. nigrum* extract, applied prior to fenitrothion treatment can be a factor increasing fenitrothion toxicity in *T. molitor* larvae. Thus, this strategy may lead to decreased emission of synthetic insecticides to the environment.

## 1. Introduction

The protection of food products imposes the rigorous usage of chemicals like insecticides, which usage worldwide increases annually. From 2011, the amount of emitted pesticides reached over 4 million tons and still increases [1]. In Poland, besides increasing usage of synthetic pesticides, the usage of insecticides from the organophosphorus group increased in the past years to 1467.4 t, where at the same time, the amount of used botanical products and biologicals decreased to only 1.1 t (data for 2017, [1]). However, considering the harmful effects of synthetic pesticides, their usage should be limited, and substitute methods such as biopesticides should be implemented. Such methods of stored products’ protection are the focus of intensive studies (for review see: [2]).

A yellow mealworm beetle (*Tenebrio molitor* L.) is considered a cosmopolitan pest, destroying grains and flour food products. Additionally, the insects pollute the food with their frass and moulted exoskeletons, making them unfit for human consumption. The insects can cause losses up to 15% of grains and flour production worldwide [3,4,5] and may act as vectors of other pests, e.g., fungi [6]. For these reasons, the yellow mealworm beetle is a model organism in many studies concerning its susceptibility to plant derivatives [7,8,9,10,11]. In previous reports, we checked the sensitivity of yellow mealworm beetle to *Solanum nigrum* fruit extract (EXT) and its main glycoalkaloids, given in the diet and in in vitro tests. We described the altered amount of biomolecules, ultrastructural malformations, increased chromatin condensation, and altered heart and oviduct contractility effects [12,13], which showed that the extract and also its main glycoalkaloids solasonine and solamargine might have affected the insect metabolism, development, and reproduction. Hence, even if natural substances may not have a lethal effect, they may significantly decrease the vitality of the insects. For the usage of *S. nigrum* extract and both glycoalkaloids speaks their impact on human health. The extract possesses medicinal properties [14] including anticancerous ones [15]. Solasonine and solamargine also have anticancer properties proved in many studies [16,17,18], having low toxicity [19,20]. Additionally, the persistence in the environment of glycoalkaloids is short [21]. Therefore, these data became a starting point to test the *S. nigrum* extract as a potential factor increasing the toxicity of synthetic insecticides. As a consequence, the amount of synthetic insecticides in use would be decreased.

For the studies, we chose fenitrothion (FN), a synthetic insecticide from the group of organophosphates, that despite its toxicity and latest withdrawal from sale by the law in European countries, is commonly used in food storage facilities in many countries worldwide. Moreover, FN is very well-studied for its chemical and physiological properties, as evidenced by numerous studies, including reports about its influence on insects [22,23,24,25], rodents [26,27,28], and humans [29,30,31]. Therefore, this pesticide is a good model for the tests of organophosphates’ toxicity. Also, the use of organophosphates, as a group of synthetic insecticides, are among those pesticides, should be limited due to their low selectivity and high toxicity to non-target animals.

The midgut is the first barrier between the consumed food and the hemocoel. It is built of a single layer of longitudinal cells covered with microvilli. Very often, the insecticidal substances enter the organism though the digestive tract with the diet—hence our attention was focused on this tissue. On the other hand, the fat body is the main tissue responsible for fat storage, important for proper endocrine physiology, development, and detoxification. Therefore, we decided to focus on these two tissues due to their importance for entrance, metabolism and storage of toxic substances in the organism. The effects of the *S. nigrum* extract in used concentrations on the midgut and fat body cells were previously described [12].

The aim of our study was to determine the influence of the EXT on FN toxicity and to find the most suitable strategy of the extract application with FN to reach the most effective results, i.e., to obtain high toxicity for the tested pest species and the low usage dose. The hypothesis to verify was: The usage of extract may increase the susceptibility of insects to FN and, possibly, decrease the amount of used FN, which causes the same effect as when only FN is used.

## 2. Results

### 2.1. Lethality of Larvae after Fenitrothion Treatment

After the first 24 h of treatment, the lethality of larvae was estimated. The values for LC_50/24h_ (lethal concentration) and LC_40/24h_ where 400 μg/mL and 300 μg/mL, respectively (Figure 1). These concentrations caused 100% mortality of larvae after 72 h of treatment and were chosen for further experiments. The lowest used concentrations (40 and 50 μg/mL) caused reversible paralysis of insects. After the application of the concentrations of 200 μg/mL and 100 μg/mL, the mortality after 72 h of treatment did not reach 100%. As a control, 70% ethanol was used, and did not show any lethal effects on larvae.

### 2.2. Lethality and Changes in the Body Mass in Larvae Treated with the Mixtures of Fenitrothion and Solanum nigrum Extract

The larvae were weighed before and after the experiment and the change of body mass (delta—∆) was calculated. The values obtained for FN and the FN–EXT blends were compared to the control, and to starving larvae. Also, the FN–EXT blends and FN-only in the concentration of 400 μg/mL, were compared. The body mass of starving larvae dropped after 3 days of starvation, but the results were not statistically significant compared to the control larvae. The difference in the body mass of larvae treated with FN compared with FN–EXT blends was not statistically significant. Highly significant statistical differences were observed between groups treated with FN and FN–EXT in relation to control and starvation treatments (Table 1.). In the previous study, the delta (∆) was calculated for larvae treated with EXT only. To compare, the values were (±SEM): 0.01%: +18.62 ± 1.28; 0.1%: +18.43 ± 1.42; 1%: +17.03 ± 1.4; 10%: +15.16 ± 1.5; *n* ≥ 53 (Spochacz et al., 2018b).

Comparing the lethality of larvae after application of FN–EXT blends to FN only, a slight, insignificant increase of lethality of larvae can be observed when the two compounds were applied. While the lethality reached 50 ± 6.9% (±SEM) after application of FN in the concentration of 400 μg/mL, the FN–EXT blends increased the mortality to 68.7 ± 6.9%, 66.6 ± 5.28% and 61.6 ± 10.2% when FN was in the mixtures with 0.1, 1 and 0.01% of the EXT, respectively. The lowest lethality of larvae after 24 h of treatment appeared after the application of the mixture of FN 400 μg/mL with the EXT at a concentration of 10% (43 ± 15.6%) (Figure 2). There was no lethality when the 70% ethanol and saline B were applied.

### 2.3. Biochemical Analysis of the Fat Body of Tenebrio molitor Larvae Treated with the Mixtures of Fenitrothion and Solanum nigrum Extract

Obtained results are presented as a percentage change in lipids, glycogen and soluble proteins content in the fat body in comparison to the control larvae (100%). The mean content of lipids in the fat body is equal to 0.74 ± 0.018 mg/mg (±SEM) of the dry mass. Despite the fact that the larvae treated with FN, as well as the insecticide and its mixtures with EXT, significantly lost weight (Table 1), the lipid content in the fat body has only slightly changed compared to the control (Figure 3A). The highest changes were observed after the application of the mixture of FN (400 μg/mL) and EXT at concentrations of 0.1 and 1%. In those variants, fat body contained about 8 ± 4.4% and 7.3 ± 3.16% more lipids than the control, respectively.

Glycogen is stored in the fat body as a source of energy. We checked its level in the fat body, and the mean content of glycogen in that tissue in control was 16.4 ± 5.95 μg/mg of dry tissue. The glycogen level decreased after the application of the FN mixtures with the lowest (0.01%) and the highest (10%) EXT concentrations (Figure 3B). The average content of glycogen after application of mentioned FN–EXT blend was lower than in the control, about 43 ± 34.3% and 38 ± 37.3%, respectively. Whereas a slight increase in glycogen content in the fat body was noted in insects treated with a mixture of FN (400 μg/mL) and EXT at a concentration of 0.1 and 1%.

After the application of FN in the concentration of 400 μg/mL and its mixtures with the EXT, the decrease in the protein level in the fat body of larvae was observed (Figure 3C). Tissue from control insects contained mean 0.08 ± 0.008 mg of proteins in 1 mg of the dry mass of tissue. The more drastic drop was observed within tissues exposed to blends, than the FN alone. The average protein content was lower at about 32 ± 10.5% and 27 ± 5.9% than the control, when the FN mixtures with the lowest (0.01%) and the highest (10%) EXT were applied, respectively.

### 2.4. The Influence on the Midgut and the Fat Body Ultrastructure

#### 2.4.1. Midgut

The final effects, observed under TEM, are presented in Figure 4 and summarized in Table A1. FN (400 μg/mL) caused an increase of electron-dense chromatin and cytoplasm, cells were rich in raw endoplasmic reticulum (RER) (Figure 4, No. 2) compared to control cells (Figure 4, No. 1). Similar effects were observed after the application of blended FN and EXT (Figure 4, No. 3–5). However, some additional changes such as swollen ER were present (Figure 4, No. 4 and 5) when FN–EXT (1 and 10%) were applied into the larvae diet.

#### 2.4.2. Determination of Chromatin Density in the Midgut Cells

The density of chromatin was calculated and presented in Figure A1. In all tested concentrations, the mean value of percentage amount increased compared to the control treated with a mixture of ethanol and saline B (mean ± SEM: 27.8 ± 1.85%). The highest increase was observed after the application of FN (400 μg/mL), which was equal to 37.3 ± 3.0%. Although clear, the changes were not statistically significant. The correlation coefficient for the first variant of treatment in the midgut cells showed moderate negative correlation: −0.65.

#### 2.4.3. Fat Body

Trophocytes present in the fat body are responsible for storing lipids in droplets, glycogen in granules deposited in the cytoplasm, and proteins in varied sizes of oval structures. Between lipid droplets filling the majority of the cell volume, a nucleus is present with the electron-dense nucleoplasm, placed in the center and close to the envelope (Figure 5, No. 1). Ultrastructural changes observed in the fat body cells treated with FN and its blends with EXT were put in order in Table A2.

Fenitrothion in the concentration of 400 μg/mL given to larvae for 3 days caused an increase of the nucleoplasm density (Figure 5, No. 2). The cytoplasm around stored proteins created electron-lucent space, which may suggest vacuolization and imbalanced osmotic conditions.

Blended FN and the EXT (0.01%) (Figure 5, No. 3) caused swelling of the intermembrane space of the nuclear envelope. What is more, in the places where the envelope was disturbed, dense nucleoplasm appeared.

A tenfold increase of the EXT concentration in the mixture with FN caused the decrease of the cytoplasm density, which could be observed together with the vacuolated areas, and fusion of the lipid droplets with their homogeneity diminished (Figure 5, No. 4). Similar effects were observed after the application of the blended FN and 1% EXT (Figure 5, No. 5). Besides the lipid droplets merging and changes in their homogeneity, part of stored proteins also showed a decrease in their homogeneity.

The highest used concentration of the FN–EXT blends caused significant disturbance of cellular membranes with no visible effects on the membranes inside the fat body cells (Figure 5, No. 6). As previously described, cytoplasm was vacuolized around stored proteins.

#### 2.4.4. Determination of Chromatin Density in the Fat Body Cells

Electron-dense nuclei observed under transmission electron microscopy suggested potential changes of the heterochromatin-electron lucent chromatin ratio. Therefore, the nuclei on the electronograms were analyzed for the amount of heterochromatin. In each used concentration, the increase of the amount of heterochromatin was observed (Figure A2) compared with control (25.0 ± 2.35%). Statistical significance was noticed in FN (400 μg/mL) and its blends with 1 and 10% EXT concentrations reaching value 35.4 ± 1.9%, 38.0 ± 1.96% and 38.9 ± 3.7%, respectively. The correlation coefficient for the first variant of treatment in the fat body cells showed a strong positive correlation: 0.72. Mortality and changes in the body mass of larvae exposed to fenitrothion pre-treated with *S. nigrum* extract.

The differences in the mass gain were calculated (Table 2). FN (300 μg/mL) caused the highest loss of the larval body mass. Both FN (300 μg/mL) and the EXT-pre-treated groups showed significant differences in the body mass compared to the control, and to starving larvae. What is more, when EXT (0.01 and 10%) was applied before FN, the body masses were significantly different from the FN-treated group. The body mass of larvae after treatment with EXT alone are presented in Section 2.2.

What seems to be the most important result in the process involving the EXT applied before FN is the significantly raised mortality of larvae (Figure 6). The *S. nigrum* extract in concentrations 0.01, 0.1, 1 and 10% increased mortality from 40 to 100, 90, 79 and 96%, respectively.

### 2.5. Biochemical Analysis of the Fat Body of Tenebrio molitor Larvae after Application of the Solanum nigrum Extract as a Preceding Factor before Fenitrothion Application

The biochemical analyses were conducted similarly as in the first variant of the experiment. Three main types of stored substances such as lipids, glycogen and soluble proteins were isolated and their concentration in the dry mass of tissue was measured. The content of lipids in the fat body after 3 days of treatment with the EXT and FN decreased significantly (Figure 7A) compared to the control (0.63 ± 0.08 mg/mg, ±SEM) but in the case of FN in the concentration of 300 μg/mL, the level of lipids scarcely, insignificantly increased (0.68 ± 0.05 mg/mg). The highest decrease was observed when the EXT in the concentration of 1% was applied before FN (0.42 ± 0.02 mg/mg). In each variant with using the EXT as a preceding factor, the average amount of lipid in the fat body was significantly lower (*** *p* < 0.001) from FN in the concentration of 300 μg/mL.

The decrease in the level of glycogen in the fat body was significant after application of all tested compounds (Figure 7B). In the control larvae, after 3 days of the experiment, the average level of glycogen in the fat body was 29.9 ± 6.15 μg/mg (±SEM). The highest decrease in the glycogen level was observed in the larvae treated with FN (300 μg/mL) and larvae treated with 0.1% EXT and FN, where glycogen level dropped to 1.5 ± 0.19 μg/mg and 1.6 ± 0.39 μg/mg, respectively.

The fraction of soluble proteins in the control larvae was 0.09 ± 0.007 mg/mg (±SEM) of dry tissue mass. The significant increase of protein (0.12 ± 0.01 mg/mg) in the fat body was observed in larvae treated with FN (300 μg/mL) (Figure 7C). It also increased after the treatment with the extract in the 10% concentration, which was 0.11 ± 0.01 mg/mg.

### 2.6. The Influence on the Midgut and the Fat Body Ultrastructure

#### 2.6.1. Midgut

The ultrastructure of midgut cells showed electron-dense cytoplasm, an increase of the amount of RER in the cytoplasm (Figure 8, No. 6), and its swelling after application of all used concentrations compared to control (Figure 8, No. 1 and 2). The results are summarized in Table A3. Cellular membranes were significantly separated in the apical part of the cell (Figure 8, No. 4 and 5) and an increase of electron-dense chromatin was observed in this variant with pre-treating with EXT (Figure 8, No. 3).

#### 2.6.2. Determination of Chromatin Density in the Midgut Cells

In the nuclei of midgut cells treated with FN (300 μg/mL) the amount of heterochromatin decreased (23.7 ± 1.69%) compared to the control (27.8 ± 1.85%) (Figure A3). The highest mean values were observed under the influence of FN with pre-treatment of 0.1% EXT (30.7 ± 2.0%). The correlation coefficient for the second variant of treatment in the midgut cells exhibited no signs of correlation: −0.18.

#### 2.6.3. Fat body

Cells of the fat body treated with FN (300 μg/mL) showed abnormalities of the cellular membranes adhesion, increased of the cytoplasm and chromatin electron-density as well as changes in homogeneity of stored proteins and lipid droplets (Figure 9, No. 3) compared to control cells (Figure 9, No. 1 and 2). When EXT was applied as a preceding factor before FN (Figure 9 No. 4–6), similar effects were observed, but with different intensity (Table A4).

#### 2.6.4. Determination of Chromatin Density in the Fat Body Cells

The amount of heterochromatin (±SEM) showed an increasing tendency after application of each concentration compared to the control (25.08 ± 7.42%) (Figure A4). The highest difference from control was observed after application of EXT in the concentration of 10% in the pre-treatment (35.2 ± 2.13%), nevertheless, noticed differences were not statistically significant. In the fat body cells treated with FN with EXT pre-treatment, the correlation coefficient showed strong positive correlation: 0.7.

### 2.7. Discussion

The first strategy of application assumed that the EXT added to the FN may increase the lethality of *T. molitor* larvae by additive action on the tissues crucial for the absorption and detoxification processes, such as fat body and midgut. What is more, we planned to mimic agrochemical application that would not last for more than two-to-three days. Therefore, we focused on the first results and acute toxicity effects. However, the larvae did not show any significant increase in mortality after 24 h of treatment. In addition, in the highest concentration of EXT (10%) mixed with FN, a slight decrease in mortality was observed (Figure 2). Due to the observed effects, the strategy of application has been changed and the EXT in four concentrations was given to the nourishment for 24 h before application of FN in the concentration of 300 μg/mL, which was a concentration causing 40% of lethality of *T. molitor* larvae after 24 h of treatment. The second variant of the experiment showed that the mortality of larvae increased significantly after application of FN, when the EXT was applied 24 h before (Figure 6). That may prove the increase of the sensitivity of larvae to synthetic insecticide under the influence of the EXT. In case of this strategy, the potentiation of the lethal toxicity of FN by EXT was observed.

The loss of the body mass of larvae treated with this mixture was significantly lower from the loss of the body mass of starving larvae (Table 1) which suggests high energy expenditure for the detoxification process [32,33]. Also, this proves that the lethal effect was not caused by deprived feeding. The loss of the weight by the larvae could be the result of water loss through open spiracles as a consequence of the paralysis and an increase in the metabolic rate after the insecticide exposure [34]. Similar results appeared in the second variant of the experiment.

Sublethal effects on the fat body of *T. molitor* larvae caused by *S. nigrum* extract were previously described [12]. The EXT caused a significant decrease in the glycogen level at a concentration of 0.1% and decrease of lipids content at a concentration 1% in the fat body cells. Further biochemical analyses of the levels of glycogen, lipids and proteins (Figure 3) in the first variant of the experiment did not show any significant changes compared to the control, which may be a confirmation of a short and strong neurotoxic, lethal effect caused by FN in the chosen concentration and its mixtures with the EXT, rather than other metabolic effects. It can also indicate slight changes of quantity or quality of particular compounds among tested substances that might have not been visible, due to the short period of exposure. Perhaps, the changes would be significant after longer exposure. However, such a strategy of application would not mimic the mode of agrochemical treatments we wanted to follow. This is confirmed by the work of Wojciechowska et al. [35], where the composition of the fat body changed in terms of the quality of compounds such as amino acids, fatty acids, cholesterol and carboxylic acids, in larvae treated with various types of insecticides. However, in our studies, particular fractions of lipids were not analysed. In the second variant of the experiment, the comparison of the amount of lipids after application of FN (300 μg/mL) and FN (300 μg/mL) with pre-treatment of EXT gave significant differences. The additive effect can be confirmed by the previous observations of lipid content in the fat body after the EXT application [12]. In this case, the acute toxic effect was postponed, and some detoxification mechanisms could appear. Additionally, lipids are used as the first energetic substrates [32] in cases such as intoxication.

Glycogen levels varied widely in the first variant of the experiment, which can be explained not only by the different state of food intake by each larvae [36], but also by the acute toxicity, which was tested in our experiment, did not allow for the development of the extensive detoxifying strategy. Perhaps, that is the reason why the observed alterations were not statistically significant. Possibly, larvae might have used energy for detoxification from another source, such as trehalose from hemolymph [37,38], which level was not examined in our studies. In the case of the second variant of our experiment, each used concentration decreased glycogen level significantly. FN in the sublethal and lethal doses given in the diet of the silkworm *Bombyx mori* was proven to decrease the glycogen level in the fat body [37] which confirms that this compound can change the energetic metabolism in insects.

Protein level in the fat body increased notably after feeding the larvae with FN (300 μg/mL). Other research reports decrease of protein level after application of FN to *B. mori* [39] and increase when insecticides from other groups and Deodar oil were applied to *T. molitor* [10]. These insects may use different compensatory mechanisms in response to stress, also, different used substances and strategies of their application may develop different reaction.

In our previous study, the EXT in the concentration of 0.01% given for 3 days to larvae, did not show any visible changes in the midgut cells ultrastructure. The concentration of 1% was responsible for the disruption of nuclear membranes and 10% EXT additionally for the appearance of glycogen vacuoles, a sign of changes in its metabolism in the midgut cells [12]. The above-mentioned observations were neither visible after the application of FN and FN–EXT blends nor in the midgut cells from larvae pre-treated with the EXT for 24 h and then FN for 48 h. Possibly, the exposure on the EXT was to short and as a result of an acute toxic action of FN, EXT did not penetrate the cells. However, this proposal is difficult to agree with, because the effects of FN–EXT blends were present in the fat body, which points that both glycoalkaloids and FN or their metabolites had to cross the midgut epithelium and reach the fat body causing malformations. Similar results were observed by Büyükgüzel et al. [40], where boric acid caused much more significant alterations of trophocytes of the fat body, than of the epithelial cells of the midgut.

In the case of midgut cells in the first variant of the experiment, neither synergy, nor additive effect between FN and EXT were observed (Table A1). The calculation of the correlation coefficient in the midgut cells for determination of chromatin density (Figure A1) showed moderate negative correlation which suggests the possible inhibition of FN activity by EXT in these cells. Midgut cells also exhibited electron-dense cytoplasm and abundance of RER, where in the two highest concentrations its swelling was observed (Table A1, Figure 4). However, in the second variant, different effects appeared, such as cellular membrane disruption. Cells were also rich in RER in all tested concentrations, and the swelling of ER was present (Table A3). This can be explained by the fact, that synthetic insecticides can cause disturbance of ER membranes and cell membranes in the midgut cells as well as mentioned chromatin condensation [41]. Both synthetic and bio-insecticides are responsible for oxidative and nitrosative stress [42], which can result in observed effects.

Effects observed in the fat body cells obtained from insects treated only with the *S. nigrum* extract were limited to the decreased cytoplasmic density and the homogeneity of lipid droplets and an increased electron-density of nuclei. Additionally, solasonine and solamargine were responsible for disturbance of lipid droplets and stored proteins [12]. The effects of cell disruption were observed in the ultrastructure of the fat body (Figure 5, Table A2) in both variants of experiment. The mixtures of *S. nigrum* extracts caused additional effects such as fusion and loss of homogeneity of lipid droplets (Figure 5, No. 4,5) suggesting some additive effect. What is interesting, these changes were not noted in the lowest and the highest used concentrations of the extract in mixtures. The fat body cells of larvae treated with EXT and then FN (Table A4; Figure 9) showed different signs of intoxication, such as vacuolization of cytoplasm around mitochondria, stored proteins, and endoplasmic reticulum, which can be symptoms of disruption of cellular membranes caused by the EXT.

Most found nuclei in the fat body cells, and also midgut cells from larvae treated with FN mixed with the EXT featured with electron-dense chromatin. Observed effects can be a sign of the response of insects to contact with xenobiotics, and may indicate the early beginning of pyknotic processes, which are characterized by shrunken and electron-dense nuclei [43,44]. The ultrastructural malformations of fat droplets are most probably due to high lipophilic properties of FN. Besides, lipids are the first reserves, used before glycogen or other energy-rich substrates [32]. Strong positive correlation obtained in the chromatin density of fat body cells treated with FN–EXT blends, suggest the increase of FN toxicity caused by increasing concentration of the EXT, especially well visible in the highest used concentration of EXT in blends (Figure 5, No. 6).

The malformation of biological membranes caused by glycoalkaloids (solasonine and solamargine) has been previously described [45]. They cause easier penetration of toxic substances to the cell [46], which could be an additive factor to increase FN toxicity. The changes in the chromatin condensation in the fat body after application of FN and with pre-treatment of EXT also suggests a potentiation effect (Figure A3), confirmed by calculated correlation coefficient, which showed strong positive correlation. Also, some pure glycoalkaloids induce liposome disruption and hemolysis [47]. Therefore, the observed malformations of cellular membranes and lipid droplets could be observed in our research. Next, FN may increase polarity and lead to enhanced water penetration [48]. In consequence, osmolarity of cytoplasm, shrinkage or swelling of intramembranous space of membrane-bound organelles may be observed.

Abovementioned observations suggest (1) relatively fast and unharmful transfer of toxic substances such as FN that is a small, non-polar molecule and is able to migrate through the cell membrane or by the paracellular pathway [49], and (2) their longer persistence in the fat body, which is the main detoxifying organ in larvae. Besides, due to high lipid content, it enables a high accumulation of toxins within fat body cells. This manifests in the disturbance of cellular and nuclear membranes, and an increase of the chromatin and cytoplasm electro-density, which next can lead to further changes in metabolism [50].

In several studies, plant-derived substances were used in binary mixtures with synthetic insecticides on insects due to increase of toxic properties, mainly by extension of mode of action on different targets obtaining synergistic effects. In many studies, the application of both plant-derived and synthetic insecticide with obtained synergism or additivity, were performed without creating their mixtures before the application [51,52]. First, we focused on a more practical aspect, where the mixtures were created to avoid multiplication of the field application. A similar strategy was used in studies of Maurya et al. [53], where imidacloprid mixed with crude petroleum ether *Ocimum basilicum* leave extract in ratio 1:1 showed synergistic properties against *Anopheles stephensi* than imidacloprid applied alone. Fenitrothion blended in ratio 1:1 with *Callitris glaucophylla, Daucus carota* or *Khaya senegalensis* extract was effective against *Culex annulirostris* [54]. Note that in both cases, FN and LC_25_ extracts were calculated and the final summary of effectiveness of both used substances could be predicted and compared with results. In our studies, we used EXT, which did not cause lethal effects [12], which could result in no increase in lethality in the first variant of the experiment. However, other studies of Maliszewska and Tęgowska [55] shows that capsaicin which does not have insecticidal properties, added to an organophosphorus insecticide, methidathion, increased its toxicity. Therefore, the effects may be alkaloid-specific. We decided to perform the second variant of the experiment and avoid blending used chemicals, which resulted in a massive increase of larvae lethality, proving the potentiation—the effect when non-lethal substance increases lethality of a toxic compound. A similar strategy was used on a mosquito (*Aedes aegypti*) with 4 h exposition on essential oils vapors before deltamethrin topical application, which also resulted in toxicity growth [56]. The effects prove that the strategy of application may significantly affect results of the agrochemical strategies.

To sum up our experiment results, we can agree that increased lethality, significant changes in the levels of glycogen and lipids and ultrastructure malformations prove that the second variant of the experiment is more effective. We assume that the mode of action of the extract depends on its sublethal effects [12], such as the membranes malformations that increase the FN permeation. Second, the detoxication of previously applied EXT may decrease the energy resources necessary for FN detoxication. And last, observed changes in the lipids stored in the fat body may decrease the ability to store the lipophilic xenobiotics (such as FN) in the fat body and their spreading in other tissues, i.e., nerve tissue.

## 3. Conclusions

Literature data and our observations suggest that the *S. nigrum* extract’s mode of action contributed to changes in the prooxidant/antioxidant balance, which disturb lipid peroxidation in stored lipid droplets, in the fat body, cellular membranes stability, and facilitated fenitrothion intoxication. According to the obtained results, *S. nigrum* extract is a potentiating factor for fenitrothion, with a subsequent 1-day application period on *T. molitor* larvae. Even though it does not cause acute lethality, it may significantly increase toxicity of fenitrothion. Pre-treatment with plant-derived products—in this case, a common plant from the Solanaceae family—can be beneficial for human health and environmental protection. The same effect can be achieved with the use of lower dose of synthetic insecticide, which had a positive effect on the decrease of its emission. Also, the observed effect can be obtained with use of relatively low concentrations of EXT. That may limit costs and perhaps the extracts can be made of plant parts which are not used by farmers as sold products. A mildly persistent natural synergist with rich composition may also prevent from increasing resistance development among insect pests. The obtained results suggest that proper strategies are crucial for increasing the insecticidal effects.

## 4. Materials and Methods

### 4.1. Materials

#### 4.1.1. Insects

*T. molitor* larvae were obtained from the breeding culture at the Department of Animal Physiology and Development. The larvae were bred under laboratory conditions at 26 °C and 60% of humidity in a 12:12 h dark to light photoperiod. For the experiments, larvae after molting with weights of 120–140 mg were used. The insect’s weight allowed to choose the larvae with a similar metabolic rate.

#### 4.1.2. Substances

Fenitrothion (Purity: 98.3 ± 0.2% (*m/m*)) was purchased from the Institute of Organic Industrial Chemistry, Annopol, Warsaw, Poland. Because of their low solubility in water, fenitrothion solutions were prepared in 70% ethanol for the lethality calculation.

The *S. nigrum* unripe berries extract was obtained from the research group of prof. Bufo, S.A. from the University of Basilicata, Potenza, Italy. The voucher specimens were deposited at the Herbarium Lucanum (HLUC, Potenza, Italy), with the ID Code: 2320. The extraction method was conducted according to the method previously described by Cataldi et al. [57]. The chemical analysis was conducted at the Department of Sciences, University of Basilicata by Prof. Sabino Bufo’s team. The extracts at concentrations of 0.01, 0.1, 1, and 10% were diluted in physiological saline B (274 mm NaCl, 19 mm KCl, 9 mm CaCl_2_).

The concentration of 70% ethanol solution of fenitrothion and *S. nigrum* extract were mixed together in the ratio 1:1 creating the final concentration of 400 μg/mL of fenitrothion and 0.01, 0.1, 1 and 10% extract solution in 10 μL of applied substance, what imitates a simultaneous application of both substances.

### 4.2. Methods

The lethal concentrations (LC_x_) of fenitrothion were calculated for *T. molitor* larvae. Solutions were prepared and applied into the nourishment for each larva. The lethality of larvae was checked after 24, 48 and 72 h of treatment. For each used concentration (40, 50, 100, 200, 300 and 400 μg/mL) a minimum of 15 larvae were used in each of the three repetitions of the experiment.

#### 4.2.1. Methods of Application

Due to the results obtained in the first variant of the experiment, the strategy of application of fenitrothion and the extract has been changed (Table 3). The *S. nigrum* extract was considered as a preceding factor that may suddenly weaken the larvae, disturb the physiology of larvae and ease the fenitrothion penetration. The extract in one of four concentrations of 0.01%, 0.1%, 1%, and 10% was given to the larvae in the first day of treatment and then fenitrothion in the concentration of 300 μg/mL was applied. The lethality was counted from the day when fenitrothion was applied and compared to lethality caused by fenitrothion in the concentration 300 μg/mL. The higher concentrations of fenitrothion (400 μg/mL) were not used due to the their strong acute effects.

a) Application of extract mixed with fenitrothion

The first method assumed the additive action of the *S. nigrum* extract and fenitrothion. The substances were mixed together in a ratio of 1:1, creating a final concentration of 400 μg/mL of fenitrothion, and the extract in one of four concentrations: 0.01, 0.1, 1, and 10%. The larvae were fed with these blends for 3 days. The mortality was noted every 24, 48 and 72 h. The results were compared with the control (saline) and the fenitrothion in the concentration of 400 μg/mL.

b) Fenitrothion application with pre-treatment of extract

Due to the results obtained in the first variant of the experiment, the strategy of application of fenitrothion and the extract has been changed (Table 3). The *S. nigrum* extract was considered as a preceding factor that may sublethally weaken the larvae, disturb the physiology of larvae and ease the fenitrothion penetration. The extract in one of four concentrations of 0.01, 0.1, 1 and 10% was given to the larvae in the first day of treatment and then fenitrothion in the concentration of 300 μg/mL was applied. The lethality was counted from the day when fenitrothion was applied and compared to lethality caused by fenitrothion in the concentration 300 μg/mL. The higher concentrations of fenitrothion (400 μg/mL) was not used due to the very strong acute effects.

c) Starvation

Additionally, a group of larvae was selected and not fed for the time of experiment due to the examination of the body mass loss during starvation. Larvae were weighed before and after the experiment, and the delta was calculated.

#### 4.2.2. Lethality and Mass Gain

The *T. molitor* larvae weighing 120–140 mg were kept separately in the glass flasks. The day after collection, insects were fed for three days with the artificial diet prepared according to David et al. [58]. Each portion of the nourishment contained 10 mL of tested substances or the solvents of the used substances such as saline (274 mm NaCl, 19 mm KCl, 9 mm CaCl_2_) or/and 70% ethanol as a control. On the fourth day, the mortality of larvae was counted, and the alive larvae were weighed and samples of the fat body and midgut were collected. The difference between the weight before and after the experiment was used to calculate the change in the body mass according to the following equation:∆ = (b × 100)/a − 100(1)
where ‘a’ is the mass of larva before and ‘b’ is the mass after the experiment.

The lethality was noted after 24, 48 and 72 h. Larvae were considered dead when the paralysis was irreversible and blocked the vital functions. For the experiments the *n* ≥ 30 individuals per concentration.

#### 4.2.3. Biochemical Analysis of the Fat Body

Small pieces (1–3 mg) of the fat body were washed with saline and placed in weighed Eppendorf tubes. All samples were dried under vacuum conditions (−0.9 atm.) at 60 °C and weighed. The content of glycogen, lipids and proteins were determined and the amount of substances was expressed as milligrams or micrograms of substance per milligram of dry mass of the tissue. Isolation and determination of glycogen, lipids and proteins content was described previously [12] for the fat body of *T. molitor* larvae treated with *S. nigrum* extract. For each analysis at least nine individuals were collected from three repetitions.

#### 4.2.4. Determination of Glycogen Level

Isolation and determination of glycogen level were carried out according to procedures of van Handel [59] and Dubois et al. [60] respectively as follows. The samples were incubated for 15 min at 90 °C with 500 μL of KOH to lyse the tissues. Then 50 μL of a saturated solution of Na_2_SO_4_ and 800 μL of 96% ethanol were added to precipitate the glycogen. The obtained suspension was centrifuged at 10,000 rpm for 10 min and the supernatant was rejected. The pellet was washed with 70% ethanol three times. After the evaporation of residual ethanol at 74 °C, 500 μL of purified water was added. The pellet was shaken for 5 min at 80 °C and then centrifuged for 5 min at 10,000 rpm. The obtained solution was used to determine the glycogen amount spectrophotometrically (Eppendorf BioSpectrometer, Hamburg, Germany). As a standard oyster glycogen (Sigma-Aldrich, St. Louis, MO, USA) was used.

#### 4.2.5. Determination of the Lipid Content

The isolation of the fat body lipids was conducted according to the Folch et al. [61] method. Dry tissues were homogenized in 1000 μL of chloroform-methanol mixture (2:1 *v/v*) and centrifuged at 10,000 rpm for 10 min. The supernatant was transferred to a new Eppendorf tube and washed three times with 220 μL of 0.29% NaCl. The remaining solution was evaporated at 30 °C under vacuum (−0.9 atm). The pellet was dissolved in 1000 μL of chloroform-methanol mixture (2:1 *v/v*) and 500 μL of the solution was transferred to the new weighed Eppendorf tubes. After drying under vacuum (30 °C, −0.9 atm), the mass of the residual lipids was measured gravimetrically and counted as milligrams of lipids in a milligram of dry mass.

#### 4.2.6. Determination of the Soluble Protein Content

Dry samples were mixed with 200 μL of saline and homogenized on ice. Samples were centrifuged at 10,000 rpm for 5 min at 4 °C. Next, 2 μL of the infranatant was placed on the PTFT membranes and measured with a Direct Detect^®^ Infrared Spectrometer (Merck Millipore, Burlington, MA, USA).

#### 4.2.7. Transmission Electron Microscopy

The larvae chosen randomly were anesthetized with carbon dioxide. Pieces of the fat body and midgut were washed with saline and fixed in 2% glutaraldehyde. The procedure was carried out according to Adamski et al. [62]. Pieces of the fat body and cleaned midgut were placed in 2% glutaraldehyde in 0.175 M cacodylate buffer for 2 h and postfixed with 2% osmium tetroxide for 2 h. Samples were subsequently dehydrated with increasing concentrations of ethanol and then embedded in Spurr Low—Viscosity Embedding Media (Polysciences, Inc., Warrington, PA, USA). Ultrathin sections of the resin were cut with a Leica ultramicrotome and stained with uranyl acetate and lead citrate. Samples were observed under JEOL 1200EX II JEM (JEOL, Tokyo, Japan) transmission electron microscope. For each analysis of the fat body and the midgut, three insects were used. The results presentation contains the most representative, exemplary electronograms with visible changes. The summary of observed changes were graded (+ slight changes, ++ intense changes, - absent) and placed in tables.

#### 4.2.8. Determination of Chromatin Density

The chromatin ratio was calculated using the stereological method counting the electron dense surface to whole nucleus surface with the computer program STEPanizer [63]. The digital grids (1024 squares pre-picture) were plotted on the images of nuclei (at least 10 per each concentration). The number of squares over the electron dense and electron lucent chromatin were counted and the ratio was calculated.

Next, the Pearson correlation coefficient concerning the concentration of the tested substances and the heterochromatin ratio was calculated. Values between −0.3 and 0.3 were regarded as having no linear relationship, values between 0.3 < × ≤ 0.5 and −0.3 > × ≥ −0.5 indicated a weak (positive/negative) relationship, values between 0.5 < × ≤ 0.7 and −0.5 > × ≥ −0.7 were regarded as having a moderate (positive/negative) relationship, values between 0.7 < × ≤ 0.9 and −0.7 > × ≥ −0.9 indicated a strong (positive/negative) relationship, and values between 0.9 < × ≤ 1 and −0.9 > × ≥ −1 indicated a full (positive/negative) relationship.

#### 4.2.9. Statistical Analysis

All the data are presented as the mean values ±SEM of n number of replicates. The statistical significance of differences between the control and treatment values was determined using statistical tests: one-way ANOVA Tukey’s test and Dunnet’s Test, or if there was not a normal distribution, the nonparametric Kruskal–Wallis test and Dunn’s Multiple Comparison Test. The Mantel–Cox test was used to compare lethality. The statistical analyses were conducted using GraphPad Prism 5 software (GraphPad Software Inc, Version 5.01, MacKiev, La Jolla, CA, USA, 1992–2007).

## Figures and Tables

**Figure 1 toxins-12-00612-f001:**
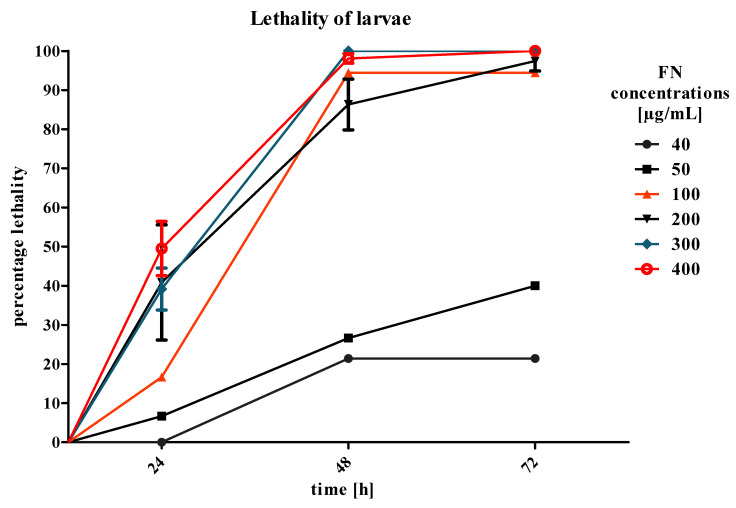
Lethality of *T. molitor* larvae (±standard error of measurement (SEM)) after application of fenitrothion (FN) in the diet in 6 increasing concentrations 40, 50, 100, 200, 300 and 400 μg/mL, *n* ≥ 15 per treatment.

**Figure 2 toxins-12-00612-f002:**
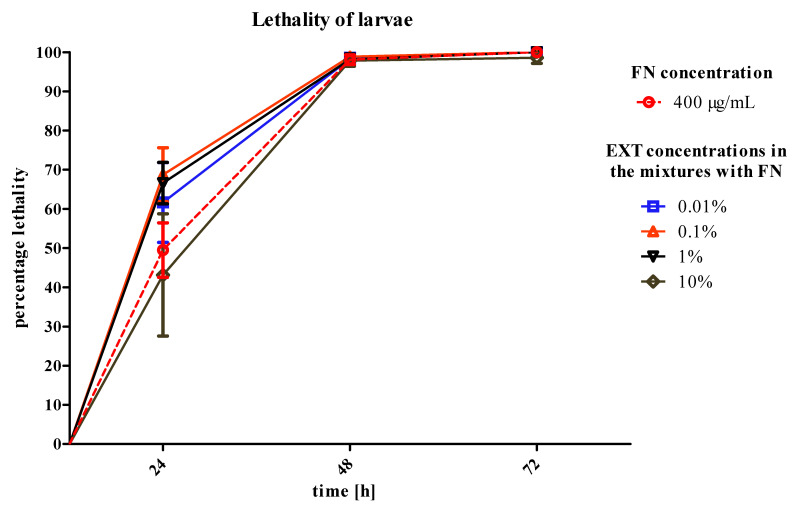
Lethality of *T. molitor* larvae after application of fenitrothion (FN) in the concentration of 400 μg/mL and its blends with *S. nigrum* extract (EXT) in four concentrations. Mantel–Cox test, *n* ≥ 46 per treatment. Changes were not statistically significant compared to fenitrothion (400 μg/mL).

**Figure 3 toxins-12-00612-f003:**
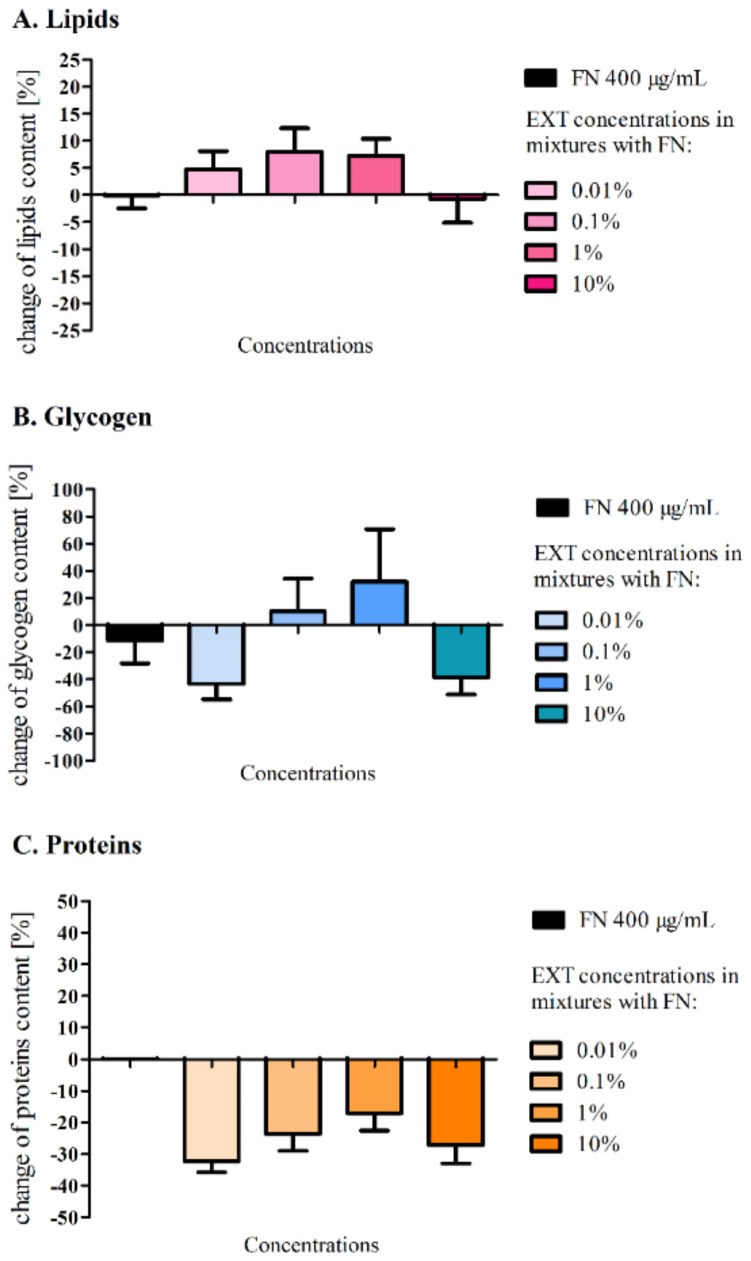
The content of lipids (**A**), glycogen (**B**) and soluble proteins (**C**) in the fat body of *T. molitor* larvae after the treatment with fenitrothion (FN) in the concentration of 400 μg/mL and its mixtures in ratio 1:1 with *S. nigrum* extract (EXT) as percentage change compared to the control larvae. Dunn’s Multiple Comparison test, *n* ≥ 9 per treatment. Changes were not statistically significant.

**Figure 4 toxins-12-00612-f004:**
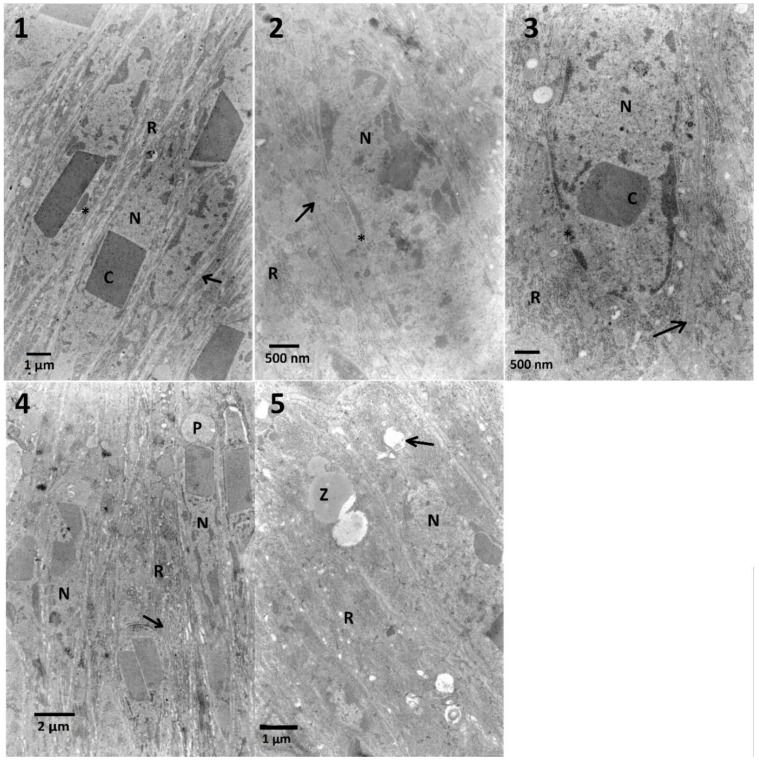
The ultrastructure of the midgut cells of *T. molitor* larvae after the treatment of the first variant, mixtures of FN (400 μg/mL) and EXT. **No. 1**—Control cells: nuclei (N) with protein crystal (C) are present. The presence of the protein crystals in the midgut nuclei is a specific feature of this species. Endoplasmic reticulum (R) surrounded by cytoplasm. Regular cell membrane and nuclear membrane are marked with an arrow and an asterisk, respectively. **No. 2**—Larvae fed with FN (400 μg/mL): an increase of cytoplasm density can be observed with an increase of ER (R), no irregularity in cellular and nuclear membranes can be observed (arrow and asterisk, respectively). **No. 3**—Larvae fed with blended FN and EXT in the concentration of 0.1%: dense cytoplasm with high amount of ER (R). An arrow points the cellular membrane. **No. 4**—1%: nuclei (N) with increased amount of dense chromatin, swollen ER (R, arrow) and undefined structures (P) present next to the nuclei, possibly vacuoles indicating beginning of cell degeneration, necrosis. **No. 5**—10%: beside swollen ER (R, arrow) in the cytoplasm other electron-dense and electron-lucent structures can be observed (Z).

**Figure 5 toxins-12-00612-f005:**
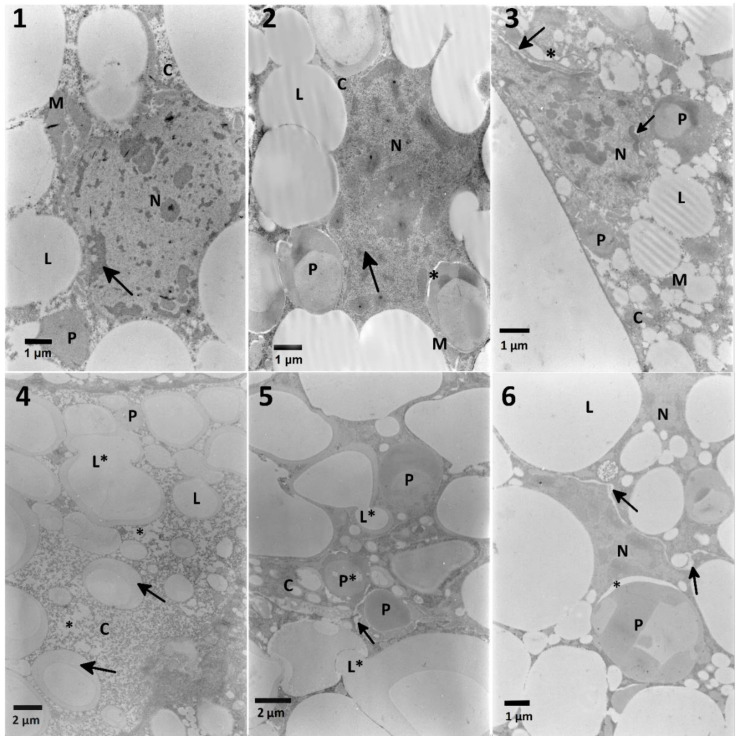
The ultrastructure of the fat body cells of *T. molitor* larvae. **No. 1**—Control cells: the regular nucleus (N) with electron-dense nucleoplasm located peripherally (arrow), dense cytoplasm with granules of glycogen (C), lipid droplets (L), stored proteins (P) and regular mitochondria (M). **No. 2**—The fat body cells of larvae treated with FN (400 μg/mL). Note an increased condensation of chromatin with large nucleolus in the center of the nucleus (arrow) and appearance of electron-lucent space between the cytoplasm and stored proteins (asterisk), and cytoplasm and mitochondria (M). The density of the cytoplasm is increased (C). **No. 3**—The fat body cells of larvae treated with the mixture of EXT (0.01%) and FN (400 μg/mL). Note that the swelling of the intermembraneous space of nuclear envelope (arrows) can be observed. **No. 4**—0.1%: changes in the lipid droplets homogeneity (arrows), the fusion of the droplets (L*) and the decrease of the cytoplasm density (C and asterisk); **No. 5**—1%: stored proteins are degraded (P*). Between cells, the increased intercellular space can be observed (arrow). The highest concentration of the EXT mixed with FN caused even more prominent increase of the intracellular space, disruption of the cellular membranes (arrows); **No. 6**—10%: note increased electron-dense chromatin within nuclei (N), electron-lucent space between stored protein (asterisk) and between cells (arrows).

**Figure 6 toxins-12-00612-f006:**
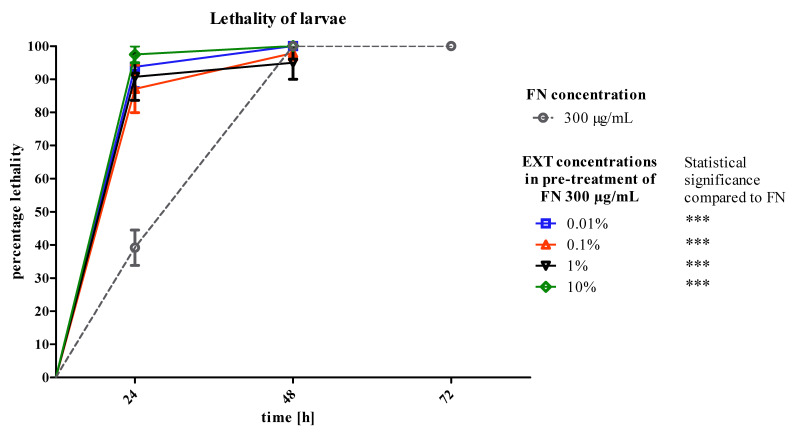
Lethality of *T. molitor* larvae after pre-treatment of *S. nigrum* EXT in four concentrations in the first day of treatment and FN in the concentration of 300 μg/mL in the second and third day of treatment compared to FN in the concentration of 300 μg/mL. Mantel–Cox test, *** *p* ≤ 0.01, *n* ≥ 30 per treatment.

**Figure 7 toxins-12-00612-f007:**
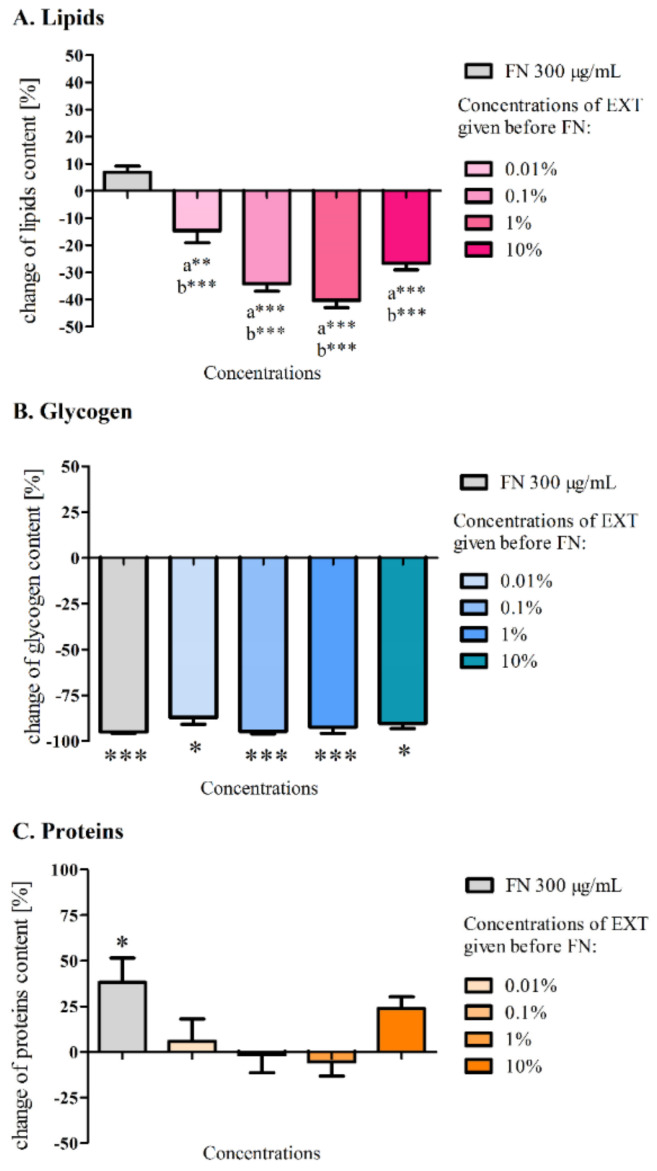
Content of lipids (**A**), glycogen (**B**) and soluble proteins (**C**) in the fat body of *T. molitor* larvae with FN (300 μg/mL) that were pre-treated with *S. nigrum* EXT in the concentrations of 0.01%. 0.1, 1 and 10% one day before FN treating. The values were compared to control and to FN with Dunn’s Multiple Comparison Test (**A**,**C**) or Dunnet’s Multiple Comparison Test (B), * *p* ≤ 0.05, ** *p* ≤ 0.1, *** *p* ≤ 0.01 compared to control (**B**,**C**) or in case of lipids (**A**) significant changes were obtained not only compared to control (a) but also to FN (b),*n* ≥ 10 per treatment.

**Figure 8 toxins-12-00612-f008:**
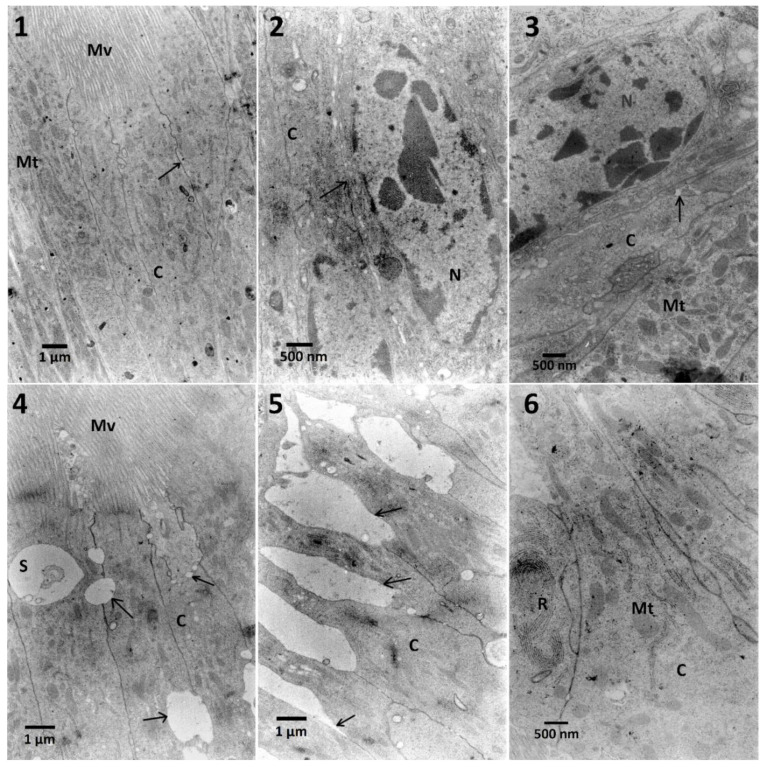
The ultrastructure of midgut cells of *T. molitor* larvae. **No. 1**—Control cells with marked microvilli (Mv), cytoplasm (C), and adherent cell membrane (arrow). **No. 2**—Midgut cell of larvae fed with FN (300 μg/mL). Nucleus (N) and cell membrane (arrow) are marked. Cytoplasm (C) increased its density. **No. 3**—Midgut cells of larvae treated with FN (300 μg/mL) with pre-treatment of 0.01% extract. Cytoplasm (C) and chromatin in the nucleus (N) are electron-dense, free space between cell membranes (arrow), and numerous mitochondria (Mt) can be observed. **No. 4**—Midgut cells of larvae treated with FN (300 μg/mL) with pre-treatment of 1% extract. Beside the disruption of cellular membranes (arrows) close to microvilli (Mv), the swelling of intermembrane space can be observed (S). The density of cytoplasm (C) increased. **No. 5**—Midgut cells of larvae treated with FN (300 μg/mL) with pre-treatment of 1% extract. In the apical part of the cells the disruption of cellular membrane is significant (arrows). **No. 6**—Midgut cells of larvae treated with FN (300 μg/mL) with pre-treatment of 10% extract. In higher magnification, extensive ER system (R) is present with mitochondria (Mt) in electro-dense cytoplasm (C).

**Figure 9 toxins-12-00612-f009:**
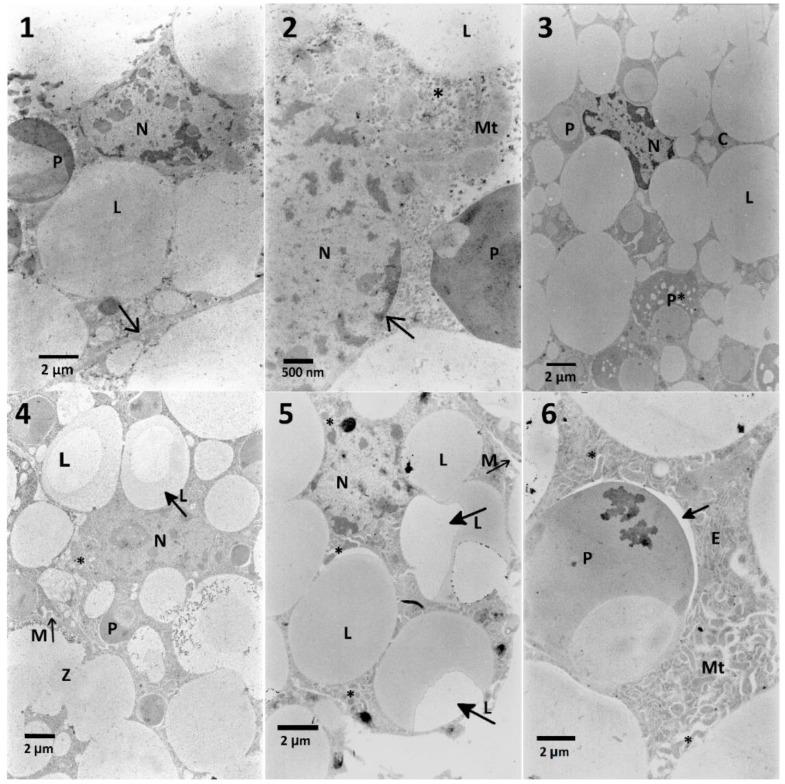
Ultrastructure of the fat body cells of *T. molitor* larvae after treatment with FN (300 μg/mL) and pre-treatment with the *S. nigrum* EXT. **No. 1,2**—Control cells. Nuclei (N) with nuclear membrane (2, arrow), lipid droplets (L), stored proteins (P) in the cytoplasm with glycogen granules (2, asterisk) with mitochondria (Mt) are visible. Cell membrane is marked with arrow (1). **No. 3**—Exemplary fat body cell treated with FN (300 μg/mL). In the center, nucleus (N) with electron-dense chromatin is present, electron-dense cytoplasm (C) and stored proteins that starts degraded (P*). **No. 4**—Pre-treatment of the lowest concentration of the EXT (0.01%) caused changes in the homogeneity of lipid droplets (arrow L) and their fusion (Z), vacuolization and decreased cytoplasmic electron density around stored proteins (P), and mitochondria (asterisk), increased density of chromatin in the nucleus (N) and disturbed cellular membranes (arrow M). **No. 5**—Inhomogeneous and irregular lipid droplets (arrows L) and mitochondria (asterisk) and disturbed cell membrane (arrow M) visible after application of 1% EXT before FN. **No. 6**—Pre-treatment of the highest concentration of the EXT (10%) caused vacuolization of the cytoplasm surrounding stored protein granules (arrow), mitochondria (M), and what is more, the distention of the endoplasmic reticulum (E).

**Table 1 toxins-12-00612-t001:** Changes in the body mass of *T. molitor* larvae after application of blended FN (400 μg/mL) with the *S. nigrum* extract (EXT).

∆ Delta, Mean Value ± SEM		Control	Starvation	FN (400 μg/mL)	FN (400 μg/mL) + EXT 0.01%	FN (400 μg/mL) + EXT 0.1%	FN (400 μg/mL) + EXT 1%	FN (400 μg/mL) + EXT 10%
		+18.97 ± 7.53	−5.59 ± 1.04	−21.28 ± 6.11	−18.09 ± 5.46	−16.9 ± 12.35	−18.71 ± 8.33	−18.14 ± 4.23
*n*		30	20	45	23	37	36	38
Difference in rank sum, significance Compared to control Compared to starvation	a)	-	31.7 ns	142.8 ***	115.2 ***	115.1 ***	119.5 ***	112.8 ***
	b)	31.7 ns	-	111.1 ***	83.45 ***	83.39 ***	87.8 ***	81.1 ***

Dunn’s Multiple Comparison test, *** *p* < 0.001. The increased body mass marked as “+”, decreased body mass marked as “-“.

**Table 2 toxins-12-00612-t002:** The changes in the body mass of *T. molitor* larvae after pre-treatment with the S. *nigrum* EXT, prior to the fenitrothion-treatment in the concentration of 300 μg/mL, compared to the control, starving larvae and pure FN (300 μg/mL).

∆ Delta, Mean Value ±SD		Control	Starvation	FN (300 μg/mL)	EXT 0.01% + FN (300 μg/mL)	EXT 0.1% + FN (300 μg/mL)	EXT 1% + FN (300 μg/mL)	EXT 10% + FN (300 μg/mL)
		+14.13 ± 9.39	−5.59 ± 1.04	−19.83 ± 3.53	−15.21 ± 5.94	−16.28 ± 5.50	−15.34 ± 6.83	−14.95 ± 5.13
*n*		21	20	32	26	17	22	21
Studentized rage distribution (q), significance, a) Compared to control b) Compared to starvation c) Compared to FN 300 μg/mL	a)	-	15.5 ***	29.7 ***	24.56 ***	22.89 ***	23.72 ***	23.15 ***
	b)	15.5 ***	-	12.27 ***	7.94 ***	7.96 ***	7.75 ***	7.35 ***
	c)	29.7 ***	12.27 ***	-	4.3 *	ns	ns	4.27 *

Tukey’s Multiple Comparison Test, * *p* < 0.05, *** *p* < 0.001.

**Table 3 toxins-12-00612-t003:** The schematic presentation of the experiments conducted in the study.

Days of Experiment/Hours of Treatment	1/-	2/0	3/24	4/48	5/72
Variant 1	Choosing larvae 120–140 mg after moulting	mixtures 1:1 Fenitrothion 400 μg/mL and *S. nigrum* extract in four concentrations: 0.01, 0.1, 1, 10%	mixtures 1:1 Fenitrothion 400 μg/mL and *S. nigrum* extract in four concentrations: 0.01, 0.1, 1, 10%	mixtures 1:1 Fenitrothion 400 μg/mL and *S. nigrum* extract in four concentrations: 0.01, 0.1, 1, 10%	Weighing the larvae after the experiment, sampling
Variant 2		*S. nigrum* extract in four concentrations: 0.01, 0.1, 1, 10%	Fenitrothion 300 μg/mL	Fenitrothion 300 μg/mL	

“-“ defines the lack of treatment in the first day of experiment.

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
