# Peer review of "Solanum nigrum Fruit Extract Increases Toxicity of Fenitrothion—A Synthetic Insecticide, in the Mealworm Beetle Tenebrio molitor Larvae"

_toxins, 2020, doi:10.3390/toxins12100612_

Round 1

Reviewer 1 Report

This study shows that the fruit extract from Solanum nigrum enhanced the toxicity by fenitrothion against Tenebrio moliter larvae. The authors elucidate that the swollen ER membrane in the midgut using the TEM observations. And also, they observed the increased amount of heterochromatin in the fat body. They quantified the body mass, lipid content and glycogen content treated with fenitrothion in the presence of S. nigrum extract. Overall, this paper is written and summarized in a clear which meets the standards for publication, but it is my opinion that for this journal, this information, as well as more scientific interpretation of the results and more detailed methodology is required to maintain the journal's quality level. I have the following major concerns about the research and the manuscript:

  1. Some experiments lack control.

Fig. 1.  The authors should show the lethality of larvae FN using 70% ethanol in the absence of FN.

Table 1. Show the mean of EXT applications without FN at 400 µg/ml.

Fig. 2. Show the lethality using 70% ethanol and physiological saline B.

Table 2.  Show the mean of EXT applications without FN at 400 µg/ml.

Table 3.  Show the mean of EXT applications without FN at 400 µg/ml.

Fig. 8. What does it mean of control? Please clarify it.

  1. line 375-377 and conclusions.  The authors don't show the evidence of "detoxification processes" and fenitrothion intoxication, respectively.

3.The authors examined the synergistic effect by EXT provided with organophosphorus insecticide, but they did not discuss why EXT assists the strong neurotoxic of FN. Why does fenitrothion with EXT affect midgut cells?

  1. minor points: line 424. S. nigrum ---> italic format.

Confirm them throughout the manuscript.

  1. line 595. at 60 oC confirm the font style.

Confirm them throughout the manuscript including similar cases.

Author Response

We would like to thank both Reviewers for their valuable remarks, which improved our manuscript.

Response to Reviewer 1 Comments

Reviewer 1:

Point 1: Some experiments lack control.

Response 1: Each experiment had a control, where saline or ethanol was given simultaneously to larvae and lethality was observed. However part of the results was already published before in the paper Spochacz et al. 2018 “Sublethal Effects of Solanum nigrum Fruit Extract and Its Pure Glycoalkaloids on the Physiology of Tenebrio molitor (Mealworm)”, where the EXT influence was tested on larvae. We put all necessary data to the text as advised to avoid confusion. Neither saline nor ethanol caused lethality of insects.

Point 2: Fig. 1.  The authors should show the lethality of larvae FN using 70% ethanol in the absence of FN.

Response 2: In the line 82 the sentence about ethanol was added.

Point 3: Table 1. Show the mean of EXT applications without FN at 400 µg/ml.

Response 3: The data obtained for the EXT was placed in the text with the reference to our previous study. We considered placing it in the table as difficult, since we used 4 different EXT concentrations that gave 4 different mean values ±SEM. In our opinion, table is easier to follow for the readers.

Point 4: Fig. 2. Show the lethality using 70% ethanol and physiological saline B.

Response 4: The proper sentence in the text was added.

Point 5: Table 2.  Show the mean of EXT applications without FN at 400 µg/ml.

Response 5: Did you mean the ultrastructural changes caused by the EXT? The results were obtained in a different study and published in a paper Spochacz et al. 2018 “Sublethal Effects of Solanum nigrum Fruit Extract and Its Pure Glycoalkaloids on the Physiology of Tenebrio molitor (Mealworm)” which was mentioned with the reference in the Discussion.

Point 6: Table 3.  Show the mean of EXT applications without FN at 400 µg/ml.

Response 6: As mentioned above, the results were obtained in a different study and published in a paper Spochacz et al. 2018 “Sublethal Effects of Solanum nigrum Fruit Extract and Its Pure Glycoalkaloids on the Physiology of Tenebrio molitor (Mealworm)” which was mentioned with the reference in the Discussion.

Point 7: Fig. 8. What does it mean of control? Please clarify it.

Response 7: The control in this case means larvae treated with mixture of ethanol and saline B. This sentence was placed in the description in lines 183-184.

Point 8: line 375-377 and conclusions.  The authors don't show the evidence of "detoxification processes" and fenitrothion intoxication, respectively.

Response 8: The line is: The first strategy of application assumed, that the EXT added to the FN may increase the lethality of T. molitor larvae by additive action on the tissues crucial for the absorption and detoxification processes, such as fat body and midgut.

The wording “detoxification processes” used in this line is a description of the fat body function. In our study we focused on the toxic effects of fenitrothion and its mixtures with extract, not on the process of detoxification. We only assume, that this process takes place in this organ, since the synthesis of main detoxification enzymes were found in the fat body. The intoxication was proven by the studies comparing the effects from the control larvae to the ones treated with fenitrothion. The significant lethality must come from the intoxication process.

Point 9: The authors examined the synergistic effect by EXT provided with organophosphorus insecticide, but they did not discuss why EXT assists the strong neurotoxic of FN. Why does fenitrothion with EXT affect midgut cells?

Response 9: At the end of the discussion a suitable summary concerning this topic has been described.

Point 10: minor points: line 424. S. nigrum ---> italic format.

Confirm them throughout the manuscript.

Response 10: We apologize for that. The change from italics to normal fonts appeared during the transfer of the ms from word to Toxins template. The text was checked for that once again.

Point 11: line 595. at 60 oC confirm the font style.

Response 11: The text was checked and all necessary font styles have been improved.

Reviewer 2 Report

Summary

With the overall purpose of reducing the application of synthetic pesticides, this study evaluates whether the use of a natural fruit extract (Black nightshade) as a pre-treatment for an organophosphorus insecticide enhances toxicity to a coleopteran pest, Tenebrio molitor, and could therefore reduce the required amounts of synthetic compound. The experimental setup is appropriate, and a variety of cellular characteristics are evaluated in addition to lethality, making the output of the paper solid and relevant. My major suggestions are to include a more in-depth discussion of the overarching question and to streamline the results section, thereby better highlighting the most interesting and relevant findings. Also, I put forward a few recommendations to present the statistics more clearly and explain a few aspects that might not be obvious for non-specialists.

General comments

  1. I suggest to explicitly comment, either in the introduction or discussion, on whether S. nigrum fruit extracts and specifically solasonine and solamargine have any known health risks for human consumption, or explicitly discuss the expectations for the used concentrations in comparison to the risks posed by fenitrothion.
  2. In the discussion, there is extensive information on mechanisms of the cellular alterations. This could be synthesized and shortened, aiming for a general overview of both approaches (experiment variants) instead of going through each experiment in such detail. Importantly, potential explanations for the different effects of applying EXT beforehand are not commented in depth. In the conclusion lines 520-521, the prooxidant/antioxidant balance is mentioned for the first time, but not discussed as a putative mechanism explaining the contrasting findings. Why the cellular/tissue damage pre-disposing for a stronger impact of FN occurs only if exposed earlier is central to the paper and thus merits more detailed discussion.
  3. Also in relation to the discussion: the 10% EXT concentration seems to repeatedly invert the dose-dependent trend observed in the lower concentrations for multiple different measurements (Fig. 2, 3, 4, 5, 13, 15, 18). Some discussion on why this could be the case and whether such effects have been observed in similar experiments seems relevant, and specially interesting from the mechanistic point of view.
  4. Please make sure that the output of statistical tests is provided for all data sets/experiments, even if non-significant (p > 0.05). This is true for many sections in the results. E.g. Fig. 5: were there significant differences? If so, what was the outcome of the test?
  5. I suggest to strongly shorten the results by combining graphs as panels (e.g. Figs 1+2, Figs 3+4+5) and moving part of the results to the supplement (Figs. 8 and 11, as well as the information contained in the tables). This should make the manuscript more concise and straightforward in delivering the key findings.

Specific comments

  • Lines 26-28: Some abbreviations used in the text are not included in this list, e.g. ER and RER. Please revise.
  • Lines 93-94: please rephrase for clarity: “Highly significant statistical differences were observed between groups treated with FN and FN-EXT in relation to control and starvation treatments”. Correct?
  • Table 1: I recommend showing these results as a graph (also see general comment nr. 5)
  • Lines 99-100: was this increase significant? Please state explicitly.
  • Line 163: are these protein crystals in the nuclei expected for control cells? Please explain ein the main text what these are, if known, and their relation to the treatments.
  • Lines 236-237: which correlation coefficient does this correspond to? (i.e. what test was applied?)
  • Lines 446-447: the wording “most prominently” might be an overstatement since apparently there is no statistically significant difference between the concentrations. Please rephrase.
  • Lines 473-474: it is not clear why the authors argue that a correlation is evidence of a synergistic effect, and not an additive one. What do the authors mean by synergistic in this case? If the implication is that the effects are higher than the sum of individual effects of FN and EXT independently, then a linear correlation of the chromatin values as described here would not demonstrate synergism. If there is more experimental or previously published data supporting that this might be synergistic (i.e corresponding values for only EXT treatment), please refer to it. This reasoning applies in general for the interpretation of synergism on other measurements in the paper.
  • Line 531-532: I suggest removing this general sentence or end with a more concrete statement specifically addressing the findings/conclusions of this work.
  • Lines 593-594: please revise this sentence
  • Line 600: “…was described previously” as detailed below? Please clarify
  • Please revise the use of italics for scientific names throughout the manuscript (mostly missing for nigrum).

Author Response

We would like to thank both Reviewers for their valuable remarks, which improved our manuscript.

Response to Reviewer 2 Comments

General comments

Point 1: I suggest to explicitly comment, either in the introduction or discussion, on whether S. nigrum fruit extracts and specifically solasonine and solamargine have any known health risks for human consumption, or explicitly discuss the expectations for the used concentrations in comparison to the risks posed by fenitrothion.

Response 1: The toxicity and benefits of used substances have been wider described in the introduction.

Point 2: In the discussion, there is extensive information on mechanisms of the cellular alterations. This could be synthesized and shortened, aiming for a general overview of both approaches (experiment variants) instead of going through each experiment in such detail.

Response 2: The variants have been described and compared together in the Discussion, where the new combined text was placed to avoid confusion.

Point 3: Importantly, potential explanations for the different effects of applying EXT beforehand are not commented in depth.

Response 3: At the end of the discussion a suitable summary has been described.

Point 4: In the conclusion lines 520-521, the prooxidant/antioxidant balance is mentioned for the first time, but not discussed as a putative mechanism explaining the contrasting findings.

Response 4: The oxidative stress was mentioned before in the description of the effects on the ultrastructure of midgut cells.

Point 5: Why the cellular/tissue damage pre-disposing for a stronger impact of FN occurs only if exposed earlier is central to the paper and thus merits more detailed discussion.

Response 5: At the end of the discussion a suitable summary of the findings has been described.

Point 6: Also in relation to the discussion: the 10% EXT concentration seems to repeatedly invert the dose-dependent trend observed in the lower concentrations for multiple different measurements (Fig. 2, 3, 4, 5, 13, 15, 18). Some discussion on why this could be the case and whether such effects have been observed in similar experiments seems relevant, and specially interesting from the mechanistic point of view.

Response 6: First, we didn’t consider this finding as relevant, since the differences were not statistically significant, and we were afraid of an exaggeration in the interpretation of data where only “a trend” can be observed. However it is true, that the highest concentration of the extract causes a slight opposite effects in each examination. We believe that the highest EXT concentration has some properties that inhibit food intake, larvae consumes less of the EXT, and in consequence the toxic effect is lower. However it was not examined and it’s not possible to make any strong statement about it. This observation suggests the need for the next experiments, focusing the feeding behavior of larvae. However, this was not the topic of the present manuscript and we plan to carry out and publish these research in the future.

Point 7: Please make sure that the output of statistical tests is provided for all data sets/experiments, even if non-significant (p > 0.05). This is true for many sections in the results. E.g. Fig. 5: were there significant differences? If so, what was the outcome of the test?

Response 7: It is true, in some places we noted the lack of significance in the text, in some not. Please, find the improvements where we marked under each Figure where was necessary the information about lack of significance. In other case the significance was marked with asterisks above the columns or data.

Point 8: I suggest to strongly shorten the results by combining graphs as panels (e.g. Figs 1+2, Figs 3+4+5) and moving part of the results to the supplement (Figs. 8 and 11, as well as the information contained in the tables). This should make the manuscript more concise and straightforward in delivering the key findings.

Response 8: Figures presenting amount of heterochromatin (Fig. 8 and 11 as well as 18 and 21) and Tables summarizing the data from TEM (2, 3 and 5,6) were placed in Supplementary materials. Figures presenting biochemical changes: 3-5 and 13-15 were attached as panels. Figures presenting electronograms were attached 6 and 7; 9 and 10; 16 and 17; 19 and 20. The numbering of tables and figures has been updated across the manuscript. The article now has 9 Figures and 3 Tables: 4 Figures and 4 Tables were transferred to Suplementary materials. This, we belive made the manuscript more clear. We are very grateful for this remark.

Specific comments

Point 9: Lines 26-28: Some abbreviations used in the text are not included in this list, e.g. ER and RER. Please revise.

Response 9: Done

Point 10: Lines 93-94: please rephrase for clarity: “Highly significant statistical differences were observed between groups treated with FN and FN-EXT in relation to control and starvation treatments”. Correct?

Response 10: Yes, thank you very much.

Point 11: Table 1: I recommend showing these results as a graph (also see general comment nr. 5)

Response 11: The first idea was to create a graph with this data, but the values of delta are +19 for the control and close to -22 for the tested compounds. Taking the control data as a 0 and the data for the tested compounds as a percentage change of control, we also obtain a huge difference that also makes the data unsuitable for a clear and strightforward presentation as a graph. Additionally, data in the table are more precise, hence we decided to keep the data about body mass gain in the tables. 

Point 12:Lines 99-100: was this increase significant? Please state explicitly.

Response 12: Done

Point 13: Line 163: are these protein crystals in the nuclei expected for control cells? Please explain ein the main text what these are, if known, and their relation to the treatments.

Response 13: The protein crystals appears in the midgut nuclei of T. molitor and it is a specific feature of this species. It is know that the crystal can be a protein storage, but no data is available about their role and whether these crystals appearance is connected to external or internal factors. The sentence has been added to the description. (Ref.: Martoja, R.; Ballan-Dufrançais, C. The ultrastructure of the digestive and excretory organs. In Insect Ultrastructure; Springer: Boston, MA, USA, 1984; pp. 199–268.)

Point 14: Lines 236-237: which correlation coefficient does this correspond to? (i.e. what test was applied?)

Response 14: We are little bit confused because the above mentioned lines do not refer to the correlation. We used Pearson correlation coefficient (added to materials and methods)

Point 15: Lines 446-447: the wording “most prominently” might be an overstatement since apparently there is no statistically significant difference between the concentrations. Please rephrase.

Response 15: That is true. Done.

Point 16: Lines 473-474: it is not clear why the authors argue that a correlation is evidence of a synergistic effect, and not an additive one. What do the authors mean by synergistic in this case? If the implication is that the effects are higher than the sum of individual effects of FN and EXT independently, then a linear correlation of the chromatin values as described here would not demonstrate synergism. If there is more experimental or previously published data supporting that this might be synergistic (i.e corresponding values for only EXT treatment), please refer to it. This reasoning applies in general for the interpretation of synergism on other measurements in the paper.

Response 16: We agree with the Reviewer in this point. We use “synergism” in reference to lethal toxicity. However, due to sublehtal effects of EXT it would be more properly to use the term “potentiation” – the not-lethal compound significantly increases the toxicity of the toxic one. The appropriate information was added to the text.

Point 17: line 531-532: I suggest removing this general sentence or end with a more concrete statement specifically addressing the findings/conclusions of this work.

Response 17: The sentence has been removed.

Point 18: Lines 593-594: please revise this sentence

Response 18: Done

Point 19: Line 600: “…was described previously” as detailed below? Please clarify

Response 19: The sentence is a repetition of sentences in the next subchapter, so it has been deleted.

Point 20: Please revise the use of italics for scientific names throughout the manuscript (mostly missing fornigrum).

Response 20: The manuscript has been checked and improved.

Round 2

Reviewer 1 Report

I have no objection against the publication.